# Metformin Attenuates UVA-Induced Skin Photoaging by Suppressing Mitophagy and the PI3K/AKT/mTOR Pathway

**DOI:** 10.3390/ijms23136960

**Published:** 2022-06-23

**Authors:** Qiuyan Chen, Haiying Zhang, Yimeng Yang, Shuming Zhang, Jing Wang, Dawei Zhang, Huimei Yu

**Affiliations:** 1Key Laboratory of Pathobiology, Ministry of Education, Department of Pathophysiology, College of Basic Medical Sciences, Jilin University, Changchun 130021, China; 18864808972@163.com (Q.C.); zhanghaiy@jlu.edu.cn (H.Z.); yangym1115@163.com (Y.Y.); sdzhangshuming@163.com (S.Z.); dwzhang@jlu.edu.cn (D.Z.); 2Department of Pharmacology, School of Pharmacy, Jilin University, Changchun 130021, China; wangj19@mails.jlu.edu.cn

**Keywords:** metformin, photoaging, UVA, mitophagy, PI3K/AKT/mTOR pathway

## Abstract

Ultraviolet (UV) radiation is a major cause of photoaging that can induce DNA damage, oxidative stress, and cellular aging. Metformin (MF) can repair DNA damage, scavenge reactive oxygen species (ROS), and protect cells. However, the mechanism by which MF inhibits cell senescence in chronic skin damage induced by UVA is unclear. In this study, human foreskin fibroblasts (HFFs) treated with UVA were used as an in vitro model and UVA-induced skin photoaging in Kunming mice was used as an in vivo model to investigate the potential skin protective mechanism of MF. The results revealed that MF treatment attenuated UVA-induced cell viability, skin aging, and activation of the PI3K/AKT/mTOR signaling pathway. Furthermore, MF treatment alleviated the mitochondrial oxidative stress and decreased mitophagy. Knockdown of Parkin by siRNA increased the clearance of MF in senescent cells. The treatment of Kunming mice with MF at a dose of 10 mg/kg/day significantly reduced UVA-induced skin roughness, epidermal thinning, collagen degradation, and skin aging. In conclusion, our experimental results suggest that MF exerts anti-photoaging effects by inhibiting mitophagy and the PI3K/AKT/mTOR signaling pathway. Therefore, our study improves the current understanding of the protective mechanism of MF against photoaging.

## 1. Introduction

Cell senescence is a complex physiological process characterized by permanent proliferation arrest [1]. More and more studies have shown that the ability of senescent cells to proliferate and differentiate gradually declines, and they become physiologically non-functional [2]. As we know, skin aging is the most common manifestation of aging. In addition to physiological aging caused by the age-related decrease in cellular repair capacity, ultraviolet A (UVA) from sunlight contributes to skin aging [3]. Clinically, photoaging initially results in skin thinning followed by wrinkles, irregular pigmentation, telangiectasia, and coarse pores [4]. Skin photoaging not only hinders appearance, but it also imposes a psychological burden on patients and affects normal work and social interaction. It also has etiological links to the occurrence of benign skin tumors, “precancerous” lesions, and skin cancer [5,6].

UVA can cause skin damage by inducing reactive oxygen species’ (ROS) production, and DNA damage to activate downstream signaling and inflammatory cytokine formation [7,8]. UVA can directly induce and cause further damage to the DNA by stimulating the production of oxygen free radicals [9]. The phosphatidylinositol 3-kinases (PI3K)/AKT signaling pathway is one of the most well-characterized and important signaling pathways that is activated in response to DNA damage [10,11].

ATM, ATR, and DNA-PKcs, such as PI3K-like kinase (PIKK) family members, are activated in response to DNA damage and induce γH2AX production [12]. Then, activated PIKK family member proteins initiate the PI3K/AKT/mTOR signaling pathway to repair DNA damage, leading to cell cycle arrest [13,14]. Furthermore, excessive intracellular oxygen free radicals can damage the mitochondria and initiate mitochondrial oxidative stress [15]. This activates a serine/threonine protein kinase, PINK1, which in turn regulates the recruitment of ubiquitin E3 ligase Parkin to the mitochondrial surface to remove damaged or dysregulated mitochondria through mitophagy [16,17].

Metformin (MF), the first-line oral treatment for type 2 diabetes, has well-defined potential side effects and has a broad safety profile [18]. In addition to its hypoglycemic effects, MF can extend lifespan by targeting several nutrition-sensing, anti-aging, and immune pathways, thereby reducing oxidative stress, inflammation, and DNA damage [19,20]. MF also has positive effects on a range of age-related degenerative diseases, including diabetes, cancer, and dementia [21]. Therefore, MF can alleviate aging by reducing oxidative stress levels, by activating AMPK, and by inhibiting mTOR phosphorylation and PI3K/AKT signaling pathway [22]. However, the role of MF in the alleviation of skin photoaging remains unclear.

In this study, UVA-induced photoaged cell models and animals were used to study the effects of MF on aging cell markers, oxidative stress levels, the PI3K/AKT/mTOR signaling pathway, and mitochondrial autophagy pathway. We explored specific mechanisms of skin photoaging alleviation related to the PI3K/AKT/mTOR signaling pathway and mitochondrial autophagy. Our study may facilitate the development and application of MF to clinically suppress UVA-induced photoaging.

## 2. Results

### 2.1. UVA Can Induce Photoaging in Foreskin Dermal Fibroblasts

After UVA irradiation of human foreskin fibroblasts (HFFs), the photoaging of cells was analyzed. The MTT assay results revealed that UVA irradiation reduced cell viability (Figure 1a). SA-β-galactosidase staining was used to determine the proportion of senescent cells. The SA-β-galactosidase staining results revealed that UVA irradiation resulted in a significantly higher percentage of SA-β-galactosidase staining in HFFs than that in the control group (Figure 1b,c). Western blotting was used to detect the expression of senescence-related proteins and apoptotic proteins. Compared with the untreated cells, UVA irradiation significantly increased the expression levels of senescence-related proteins P53, P21, and P16 in HFFs, while activation of apoptotic proteins cleaved caspase-3, and the expression levels of Bcl-2 proteins Bax/Bcl-2 and Bak/Mcl-1 were not significantly changed (Figure 2d,e). To further detect HFFs apoptosis and to observe their nuclear morphology, Hoechst 33,342 staining was used. The results revealed that UVA irradiation of HFFs for three days had no significant effect on nuclear morphology (Figure 1d). Meanwhile, flow cytometry (FCM) was used to detect the HFFs’ cell cycle. The results revealed that UVA could cause cells to arrest in G2/M phase and cause cell cycle arrest (Figure 2c). These results suggested that UVA irradiation induces photoaging in HFF cells.

### 2.2. Metformin Can Ameliorate UVA-Induced Photoaging on HFFs

MTT assay results revealed that MF was not toxic to HFFs, and 100 μM MF treatment improved the survival rate of HFFs induced by UVA irradiation (Figure 2a). The effect of MF on UVA-induced cellular senescence was then examined. As shown in Figure 2b, MF treatment significantly reduced the ratio of SA-β-galactosidase staining compared to UVA-irradiated cells. The cell cycle was detected using flow cytometry. MF reduced the proportion of cells in the G2/M phase after UVA irradiation (Figure 2c). Western blotting was used to detect the protein expression levels. As shown in Figure 2d,e, MF reduced the expression levels of P53, P21 and P16 in UVA-irradiated cells. In MF-treated photoaged cells, the protein expression levels of Bax/Bcl−2, BAK/Mcl−1, and cleaved caspase−3 increased. The above results indicated that MF could improve cell photoaging induced by UVA irradiation and induce apoptosis in photoaging cells.

### 2.3. Metformin Inhibits PI3K/AKT/mTOR Signaling Pathway of HFFs

The most direct damage of ultraviolet (UV) radiation is DNA damage, resulting in DNA breakage and mismatch. In addition, UV radiation can cause the production of substantial amounts of reactive oxygen species (ROS) in cells, which further damages DNA. The PI3K/AKT/mTOR pathway is crucial for DNA damage repair, cell proliferation, apoptosis, inflammation, and other cellular activities. Protein expression in the PI3K/AKT/mTOR pathway was detected by western blotting. Figure 3 showed that the phosphorylation levels of PI3K, AKT, mTOR, and 4EBP1 were significantly enhanced in HFF cells upon UVA irradiation. However, MF treatment significantly reduced the expression levels of these proteins in photoaged HFF cells. This suggests that UVA irradiation induces the initiation of the PI3K/AKT/mTOR signaling pathway, whereas MF inhibits its activation.

### 2.4. Metformin Reduces UVA-Induced Oxidative Stress

Mitochondria are one of the main targets of ROS destruction, and severe mitochondrial oxidative damage usually leads to mitochondrial dysfunction. JC−1 was used to detect the effect of MF on mitochondrial membrane potential in UVA-induced photoaged HFFs. As shown in Figure 4a, the mitochondrial membrane potential of photoaged cells was significantly reduced by UVA irradiation, while MF treatment upregulated the mitochondrial membrane potential of photoaged cells. In addition, MitoTracker Red CMXRos was used to detect ROS levels in mitochondria. As shown in Figure 4b, compared to untreated HFFs, the mitochondrial ROS level in photoaged cells induced by UVA irradiation was significantly enhanced, while MF reduced the accumulation of ROS induced by UVA irradiation. Elevated levels of ROS in cells may activate the HIF–1α pathway; western blotting was used to detect the protein expression levels of HIF–1α, HO–1, and Nrf2. UVA induced HIF–1α, HO–1, and Nrf2 protein expression and combined treatment with MF reduced HIF−1α, HO−1, and Nrf2 protein expression levels (Figure 4c−f). These results suggest that MF can alleviate UVA-induced oxidative stress.

### 2.5. Metformin Can Reduce UVA-Induced Mitophagy in HFFs

Mitophagy is a form of autophagy that selectively targets damaged mitochondria to maintain mitochondrial homeostasis. Additionally, mitochondrial division determines mitophagy. To investigate whether UVA can activate PINK1/Parkin-dependent mitophagy and the effect of MF on mitophagy in photoaged cells, the protein expression of Drp1, PINK1, and Parkin in HFFs was detected using western blotting. Figure 5 shows that UVA irradiation significantly increased the expression levels of Drp1, PINK1, and Parkin, whereas MF decreased the expression levels of these three proteins in photoaged cells. Therefore, UVA exposure induced PINK1/Parkin-dependent mitochondrial autophagy in HFFs cells, whereas MF reduced PINK1/Parkin-dependent mitochondrial autophagy in photoaged cells.

### 2.6. Alteration of Parkin Expression Can Regulate Apoptosis in HFF Cells after UVA Irradiation

To further verify the effect of mitophagy on UVA-exposed HFF cells, we used siRNA to knockdown Parkin expression. Western blotting results revealed that si Parkin significantly downregulated the expression of Parkin without changing the expression level of PINK1 in the HFFs, indicating that mitophagy was effectively inhibited (Figure 6a–c). To determine the effect of mitophagy inhibition on the HFFs, the apoptosis of the HFFs after parkin knockdown was detected. Western blotting results revealed that after parkin knockdown, the ratio of apoptosis-related proteins Bax/Bcl−2 and Bak/Mcl−1 in UVA-irradiated HFFs increased, and MF treatment increased the ratio of Bax/Bcl−2 and Bak/Mcl-1 in photoaged cells (Figure 6d–f). These results indicate that the inhibition of mitophagy can increase apoptosis in photoaging cells, whereas MF can promote apoptosis.

### 2.7. Metformin Can Improve UVA-Induced Skin Photoaging in Mice

The effects of UVA irradiation and MF treatment on the general morphology of mice were verified. Before the beginning of the experiment, the mice in each group had a uniform and shiny coat color, normal food intake and stool, good mental state, sensitive response, and free movement. Eight weeks later, the back skin of the UVA-irradiated mice showed desquamation, redness, rough skin, and deepened lines. MF treatment is known to improve photoaging skin conditions (Figure 7a).

HE staining results revealed that the epidermis of the untreated mice was normal, with a complete structure, orderly arrangement and uniform distribution of dermal collagen fibers, normal blood vessels, and moderate cell composition and number. The epidermis and dermis of UVA-irradiated mice were thinned and accompanied by hypokeratosis, denatured and disordered dermal collagen fibers, thickened, broken, and unevenly distributed collagen bundles, telangiectasis, proliferated accessory organs, and inflammatory cell infiltration. MF combined with UVA treatment increased epidermal thickness, keratosis appeared normal, dermal collagen fiber degeneration was not clear, there was a regular arrangement, and the number and distribution of collagen bundles appeared normal. The degree and number of dilated capillaries and inflammatory cell infiltrations were reduced (Figure 7b).

In addition, western blotting was used to detect the expression level of the photoaging-related proteins MMP1 in mice skin. The results revealed that compared with the untreated mice, the MMP1 protein expression level in the mice exposed to UVA increased, and the MMP1 protein expression level in the skin of the MF combined with UVA-treated mice decreased (Figure 7c–e). The results revealed that UVA irradiation can lead to skin aging in mice, with the increased breakdown of collagen. Western blotting was used to detect the expression of the mitophagy protein Parkin. The results revealed that UVA irradiation increased the expression level of Parkin in mouse skin, while MF reduced its expression level.

## 3. Discussion

Ultraviolet radiation, abundant in the environment, is harmful to cells and organisms [23]. UVA rays are the most dominant ultraviolet rays on the Earth’s surface, and long-term exposure to UVA leads to skin photoaging [24]. UV exposure hinders cell renewal and reduces tissue regeneration and repair capabilities, resulting in skin aging, the deepening and widening of wrinkles, hyperpigmentation, and cancer in severe cases [25,26]. To maintain homeostasis and delay aging, the body must generate new cells and eliminate senescent cells [27]. Discovering drugs that eliminate senescent cells and accelerate the production of new cells is crucial to prevent and treat photoaging.

The features of cellular photoaging were consistent with those of cellular senescence and included cell cycle arrest, the upregulated expression of cell cycle inhibitors P16 and P21, absent expression of cell proliferation factor Ki67, inactivation of apoptosis-related proteins, and quiescent state of cells [28,29]. Currently, no factor can be used as the only specific marker of aging, and the internationally recognized method for the identification of aging is to combine two or more aging markers. When senescent cells are under appropriate stimulation, cell senescence can be reversed by certain conditions; however, if the damage is too heavy, it is irreversible [30]. In this study, UVA irradiation inhibited cell viability. Combined with various detection methods, we found that UVA induced photoaging of HFFs, which could be reversed by MF. On the one hand, MF not only has hypoglycemic and anti-inflammatory effects, but it also has the effect of slowing down skin photoaging. Thus, early intervention for skin photoaging caused by UVA can be reversed to a certain extent.

UV radiation can cause DNA damage, whereas ROS in cells further aggravate DNA damage [31].(Figure 8) In this study, the PI3K/AKT/mTOR pathway was activated to repair damaged DNA during UVA irradiation but was not significantly activated by MF treatment. In contrast, MF repaired damaged DNA by reducing the expression levels of γH2AX and phosphorylated ATM, whereas MF reduced DNA damage by reducing oxidative stress in cells [32]. The reduction in DNA damage inhibits the activation of PIKK, thereby inhibiting the initiation of PI3K/AKT/mTOR signaling [33].

Senescent cells are characterized by high levels of oxidative stress [34]. UVA irradiation enhanced intracellular ROS levels and decreased the mitochondrial membrane potential in photoaged cells, whereas MF treatment reduced ROS levels and increased the mitochondrial membrane potential [35].

Mitophagy is activated in senescent cells [36]. Mitochondrial depolarization is a hallmark of mitophagy [37]. Furthermore, Drp1 plays a key role in mitophagy progression. The activation of Drp1 results in structural and functional abnormalities of the mitochondria [38]. Parkin and PINK1 are two protein markers that examine mitophagy and play a key role in mitochondrial function [39]. PINK1 accumulates in damaged mitochondria and activates Parkin to promote mitophagy [40]. Western blotting showed that the expression level of Drp1 increased after UVA irradiation, and MF pretreatment decreased the expression level of Drp1. To determine the role of MF in UVA-induced photoaging cells, western blotting was used to detect the expression of PINK1 and Parkin, and these expression levels increased under UVA irradiation, whereas MF decreased the expression levels of UVA-induced PINK1 and Parkin and promoted the apoptosis of photoaging cells. To further study the role of mitophagy in photoaged cells, Parkin was knocked down by transient transfection, and MF significantly enhanced the level of apoptosis in photoaged cells. This finding may indicate that when UVA is irradiated, cells activate mitophagy to remove damaged mitochondria, reduce ROS accumulation, maintain photoaging cell homeostasis, inhibit mitophagy, disrupt photoaging cell homeostasis, and promote apoptosis.

Skin damage is mainly caused by the loss of collagen, which is manifested by a reduction in collagen synthesis and increased decomposition [41]. UVA irradiation induces collagen degradation and reduces protein synthesis. MMP1 is the main enzyme involved in skin degradation induced by ultraviolet rays [42]. The results revealed that MF caused a decrease in the expression level of MMP1 protein in mice. MF can also inhibit collagen decomposition.

In this study, we found that there was a new role of MF in alleviating chronic photoaging. This is in addition to anti-tumor and hypoglycemic effects. At present, there are many oncogenes involved in skin photoaging, such as TGFβ, KRAS, and BRAF, but their mechanism of action is still unclear [43,44]. In the future, the role of oncogenes in skin photoaging needs to be further investigated.

## 4. Materials and Methods

### 4.1. Extraction and Culture of Foreskin Fibroblasts

Primary HFFs were isolated from circumcised foreskins of healthy human donors aged from 5 to 16 years. Primary HFFs were digested with neutral protease and type I collagenase, and then treated with high-glucose DMEM medium (Gibco Life Technologies) containing 10% fetal bovine serum (Invitrogen, Carlsbad, CA, USA), 1% penicillin–streptomycin at a temperature of 37 °C in a 5% CO_2_ incubator. Primary HFFs were obtained with written consent from voluntary, informed donors, following a protocol approved by the Institutional Review Board of The First Hospital of Jilin University, Jilin University.

### 4.2. Human Foreskin Fibroblasts Photoaging Model

For the induction of senescence in fibroblasts by UVA radiation, briefly, cells were first grown to 50% confluency and then irradiated daily with UVA 5 J/cm^2^ for consecutive 3 days in phosphate-buffered saline (PBS), continued to culture for 48 h, and then detected.

### 4.3. Reagents and Antibodies

Metformin was purchased from MedChemExpress (Monmouth Junction, NJ, USA). 3-(4,5-Dimethylthiazol-2-yl)-2, 5-diphenyltetrazolium bromide (MTT) was purchased from Sigma-Aldrich (St. Louis, MO, USA). The following antibodies were used: anti-P53, anti-P21, anti-PI3K, anti-p-PI3K, anti-AKT, anti-p-AKT, anti-4EBP1, anti-p-4EBP1, anti-Bcl−2, anti-BAX, anti-cleaved caspase−3, anti-HIF−1α, anti-BAK (Cell Signaling Technology, Danvers, MA, USA); anti-P16, anti-GAPDH, anti-mTOR, anti-p-mTOR (Proteintech group, Inc., Rosemont, IL, USA); anti-Drp1, anti-Mcl−1, anti-PINK1, anti-Parkin, anti-MMP1 (Abcam, Cambridge, MA, USA); anti-HO−1, anti-Nrf2(Sangon Biotech, Shanghai, China); the antibodies were diluted according to the manufacturer’s instructions.

### 4.4. SA-β-Galactosidase Staining

SA-*β*-galactosidase activity was determined using the Senescent β-Galactosidase Staining Kit (Beyotime, Shanghai, China). Primary cells were washed twice with PBS after treatment. The cells were then fixed with 4% paraformaldehyde at room temperature for 15 min. Next, the cells were washed three times with ice-cold PBS and incubated in freshly prepared *β*-galactosidase staining solution at 37 °C free of CO_2_ incubator overnight. Stained cells were imaged using an Olympus FV1000 confocal laser microscope to evaluate the ratio of aged cells.

### 4.5. Western Blotting Analysis

HFF cells were treated with compounds according to experimental requirements. Then cells were lysed using RIPA lysis buffer containing proteasome inhibitors and incubated on ice for 45 min. The lysates were centrifuged at 4500 rpm for 10 min at 4 °C, the protein levels of supernatant were quantified with a bicinchoninic acid (BCA) kit (Thermo Fisher Scientific, Waltham, MA, USA) according to the manufacturer’s instructions. Proteins were separated on dodecyl sulfate, sodium salt-polyacrylamide gel electrophoresis (SDS-PAGE), transferred on poly-vinylidene fluoride (PVDF) membranes, and blocked with 10% skimmed milk for 1 h. The membranes were then probed with primary antibodies overnight at 4 °C, followed by secondary antibody for 2 h at room temperature. Electrochemical luminescence (ECL) reagent (Thermo Fisher Scientific, MA, USA) was used for immune detection and visualization using Syngene Bio Imaging (Synoptics, Cambridge, UK). GAPDH was used as control.

### 4.6. Nuclear Staining by Hoechst 33342

HFF cells were seeded in 6-well plates at a density of 1 × 10^5^ cells/well for 24 h. Then, the cells were treated daily with 5 J/cm^2^ UVA and 100 μM metformin for three consecutive days, both in singular and combination groups. After treatment, treated cells were collected and fixed with 4% paraformaldehyde for 15 min at room temperature, washed, and stained with 167 μmol/L Hoechst 33,258 (Beyotime, Shanghai, China) at 37 °C for 10 min. Finally, cells were observed under an Echo Lab Revolve microscope (San Diego, CA, USA).

### 4.7. Cell Viability Assay

The effect of cellular cytotoxicity on cell survival was assessed using the MTT assay [45]. HFF cells were seeded in 96-well plates at a density of 5000 cells/well. After 24 h, cells were treated with different concentrations of UVA and metformin. After treatment, MTT solution (5 mg/mL) was added to each well and incubation was carried out for 4–6 h at 37 °C. The supernatant was removed, and DMSO was added over a 5-min period on a shaker. Cell viability was measured at an absorbance of 490 nm using a Multiskan Spectrum (BioTek, Winooski, VT, USA)

### 4.8. Cell Cycle Assay

HFF cells were seeded in 6-well plates at a density of 1 × 10^5^ cells/well for 24 h. Then, the cells were treated with 5 J/cm^2^ UVA one day and 100 μM metformin for three consecutive days, both in singular and combination groups. After exposure to different treatments, cells were collected and fixed in 70% ethanol at 4 °C overnight. Then the PI staining was performed according to the manufacturer’s instructions. Finally, samples were analyzed by Accuri C6 Flow Cytometry (BD Biosciences), and their cell cycle-dependent distribution was analyzed using the ModFit LT 3.0 software.

### 4.9. Mitochondrial Membrane Potential (MMP) Assay 

Mitochondrial membrane potential was assayed using JC-1 according to manufacture’ s instruction (Beyotime, Shanghai, China). Briefly, cells were incubated with JC-1 dye at 37 °C under humidified air containing 5% CO_2_ for 20 min in the dark. Then, the cells were washed with staining buffer and centrifuged at 600× *g* for 4 min at 4 °C thrice. Finally, cells were suspended in staining buffer and analyzed using a BD Accuri C6 flow cytometer (BD Biosciences, San Jose, CA, USA).

### 4.10. Mitochondrial Reactive Oxygen Species (ROS) Detection 

The average level of ROS was measured using a mitochondrial ROS assay kit MitoTracker Red CMXRos (Beyotime Biotech, Shanghai, China). Briefly, after exposure to different treatments, cells were stained with MitoTracker Red CMXRos working solution and prewarmed (37 °C) for 20 min. After staining was complete, MitoTracker Red CMXRos working solution was removed and fresh cell culture solution pre-incubated at 37 °C was added. Finally, cells were observed under an Echo Lab Revolve microscope (San Diego, CA, USA).

### 4.11. siRNA Transient Transfection 

Small interfering RNA (siRNA) sequences targeting human Parkin (GenBank: AB009973.1) and a non-target sequence were purchased from Genechem (Shanghai, China). Transfections with siRNA were performed using LipofectAMINE 2000 (Invitrogen, Carlsbad, CA) according to manufacturer’s protocol. Briefly, HFFs were placed into 6 cm dishes, and transfected with 4 μg si Parkin or si Scramble 48 h after treatment using 10 μl (1 μg/μl) LipofectAMINE 2000. Cells were harvested 2 days after transfection; whole cell lysates were isolated for western blots. 

### 4.12. Experiment Animals and Treatment Protocols 

Six-week-old female Kunming mice, SPF grade, were obtained from Changchun Yisi Experimental Animal Technology (Changchun, China). Kunming mice were kept at a constant temperature (22 °C), with a light/dark cycle of 12 h, and were provided access to food and tap water. Briefly, dorsal skin sections from six-week-old mice were subdivided into the following four areas: (i) control group; (ii) UVA irradiation group; (iii) UVB irradiation + metformin (10 mg/kg) group; and (iv) metformin (10 mg/kg) group. The dorsal skin areas were irradiated with UVA (20 mJ/cm^2^ per exposure, 5 times a week for 8 weeks). Each group consisted of six animals. All animals used in the procedures were handled according to the Guide for the Care and Use of Laboratory Animals, and the in vivo experimental methods were approved by the University Committee in the Use of Animals of Jilin University, China.

### 4.13. Hematoxylin and Eosin (HE) Staining

All animals were weighed and injected with 1–3% isoflurane before sacrificed. After mice were sacrificed, skin tissues were dissected and fixed in formalin. Later, the skin tissues were embedded in paraffin to make the sections, which later underwent hematoxylin and eosin (HE) staining, whereas snap-frozen tissues were subjected to western blotting analyses. All procedures were approved by the University Committee in the Use of Animals of Jilin University, China.

### 4.14. Data Analysis

All results are based on at least three independent experiments. All data are given as mean values ± standard deviation (S.D). Results between the two groups are compared using Student’s *t*-test. *p* < 0.05 was considered statistically significant difference, and *p* < 0.01 was considered extremely significant. Statistical analysis was performed with GraphPad Prism 8.0 (La Jolla, CA, USA).

## 5. Conclusions

In conclusion, HFFs underwent photoaging induced by UVA irradiation. In addition, the PI3K/AKT/mTOR signaling pathway was activated to repair UVA-induced DNA damage, and the mitophagy pathway was activated to respond to UVA-induced cellular oxidative stress, which is a relatively stable state of photoaging cells. MF can improve UVA-induced photoaging by diminishing oxidative stress, inhibiting the PI3K/AKT/mTOR pathway and mitophagy, and promoting the apoptosis of aging cells. In other words, senescent cells are eliminated by apoptosis, reducing the proportion of senescent cells in the body, which in turn increases the proportion of non-senescent cells, slowing the photoaging skin tissues. This study therefore helps elucidate the photoaging mechanism of skin cells and provides a theoretical basis for the clinical development of MF as a drug for alleviating skin photoaging.

## Figures and Tables

**Figure 1 ijms-23-06960-f001:**
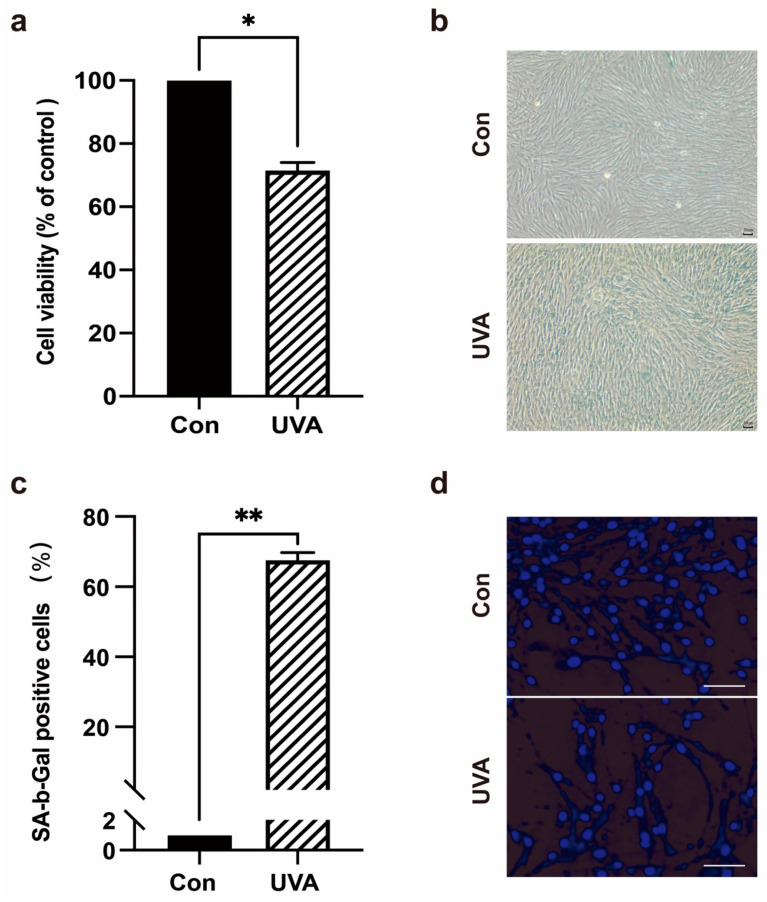
UVA can induce photoaging in foreskin dermal fibroblasts. (**a**) Cells were treated with UVA. The cell viability of HFFs were detected by MTT assay. (**b**,**c**) Cells were treated with UVA, the activity of senescence-associated β-galactosidase and the percentage of cells positive for SA-β-galactosidase staining (40×) were measured. (**d**) HFFs were treated with UVA, stained with Hoechst 33258. Cell morphology was observed by fluorescence microscopy (200×). Data are presented as mean ± SD, n = 3. * *p* < 0.05, ** *p* < 0.01.

**Figure 2 ijms-23-06960-f002:**
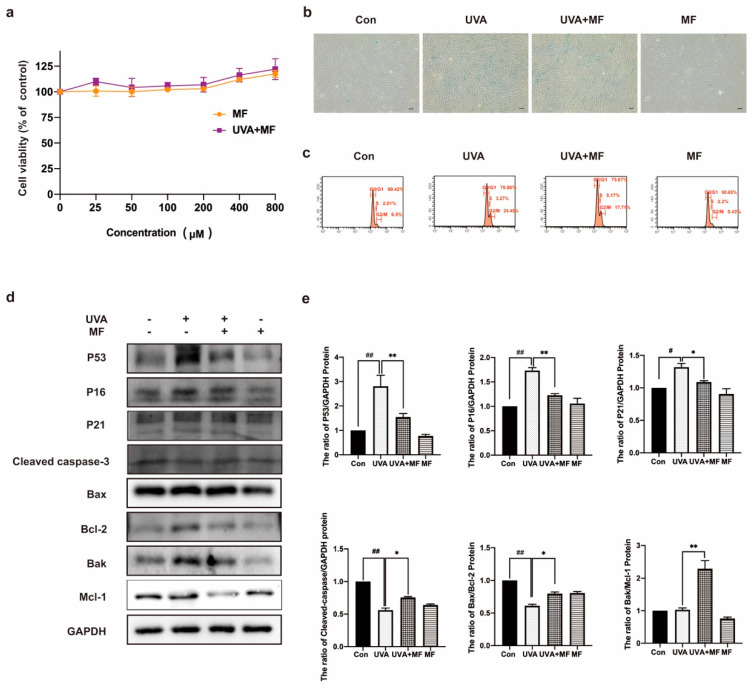
Metformin can ameliorate UVA-induced photoaging on HFFs. (**a**) HFFs were treated with metformin (0−800 μM) alone or in combination with UVA, the cell viability of HFFs were detected by MTT assay. (**b**) Cells were treated with UVA alone or in combination with 100 μM metformin, the activity of senescence-associated β-galactosidase and the percentage of cells positive for SA-β-gal activity (40×) were measured. (**c**) Cells were treated with UVA alone or in combination with 100 μM metformin, flow cytometry system analysis was used to detect the phase of the cell cycle in cells. (**d**,**e**) Cells were treated with UVA alone or in combination with 100 μM metformin, the protein levels of P53, P16, P21, cleaved caspase-3, Bax, Bcl−2, Bak, Mcl−1 were measured by western blot. Data are presented as mean ± SD, n = 3. ^#^
*p* < 0.05, ^##^
*p* < 0.01 vs. control cells, * *p* < 0.05, ** *p* < 0.01 vs. UVA-treated cells.

**Figure 3 ijms-23-06960-f003:**
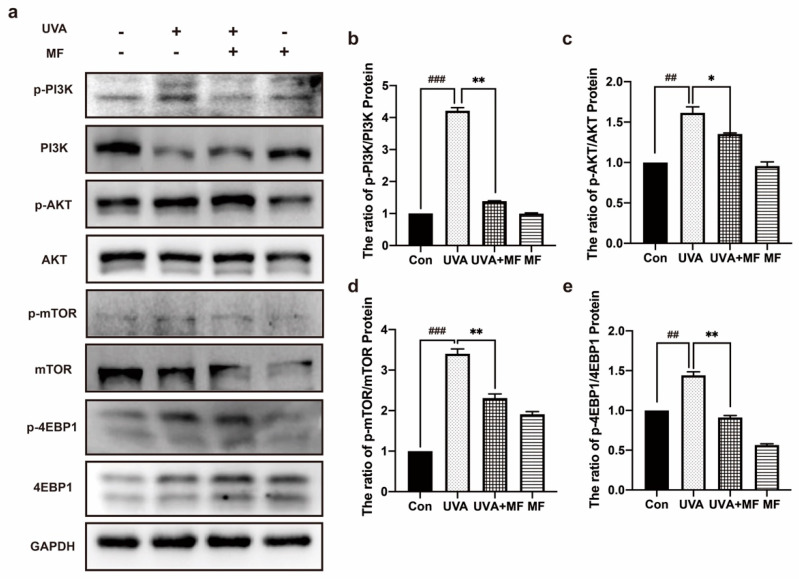
The PI3K/AKT/mTOR signaling pathway was decreased by metformin in UVA-induced senescent cells. (**a**−**e**) Cells were treated with UVA alone or in combination with 100 μM metformin, the protein levels of p-PI3K/PI3K, p-AKT/AKT, p-mTOR/mTOR, p-4EBP1/4EBP1, were measured by western blot. Data are presented as mean ± SD, n = 3. ^##^
*p* < 0.01, ^###^
*p* < 0.001 vs. control cells, * *p* < 0.05, ** *p* <0.01 vs. UVA-treated cells.

**Figure 4 ijms-23-06960-f004:**
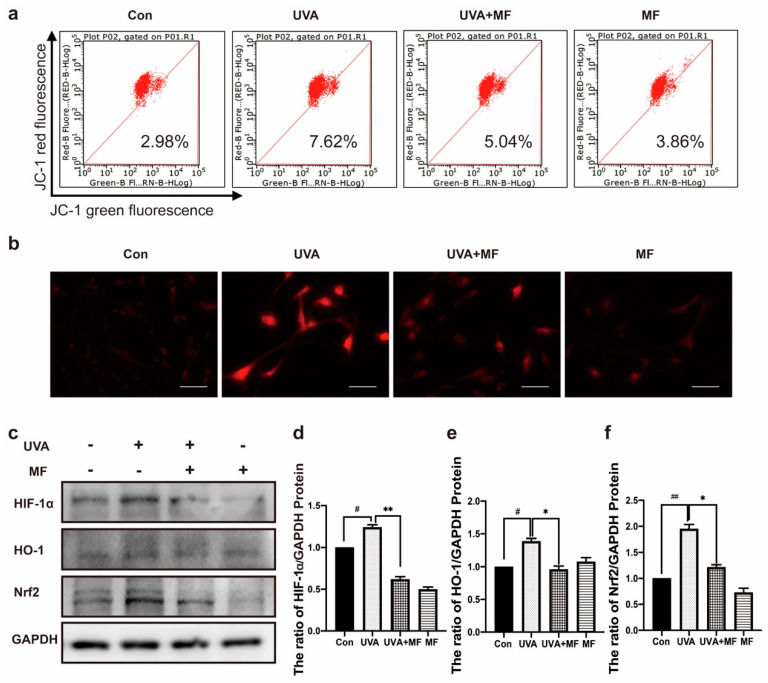
Metformin reduces UVA-induced oxidative stress. (**a**) Cells were treated with UVA alone or in combination with 100 μM metformin. Mitochondrial membrane potential was detected by flow cytometry. (**b**) Cells were treated with UVA alone or in combination with 100 μM metformin. Mitochondrial ROS generation was detected by MitoTracker Red CMXRos under fluorescence microscope (200×). (**c**−**f**) Cells were treated with UVA alone or in combination with 100 μM metformin, the protein levels of HIF−1α, HO−1, and Nrf2 were measured by western blot. Data are presented as mean ± SD, n = 3. ^#^
*p* < 0.05, ^##^
*p* < 0.01 vs. control cells, * *p* < 0.05, ** *p* <0.01 vs. UVA-treated cells.

**Figure 5 ijms-23-06960-f005:**
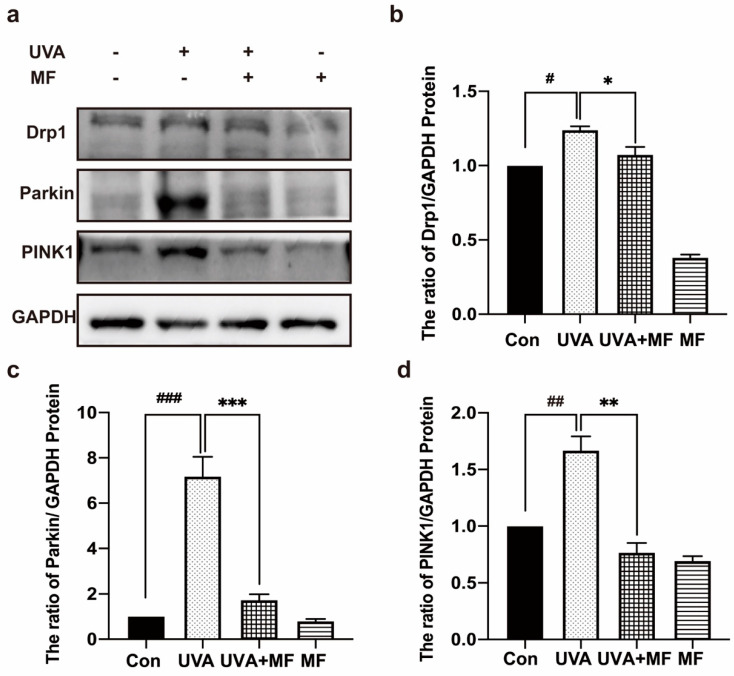
Metformin reduces UVA-induced mitophagy in HFFs. (**a**–**d**) Cells were treated with UVA alone or in combination with 100 μM metformin. The protein levels of Drp1, PINK1, Parkin were measured by western blot. Data are presented as mean ± SD, n = 3. ^#^
*p* < 0.05, ^##^
*p* < 0.01, ^###^
*p* < 0.001 vs. control cells, * *p* < 0.05, ** *p* < 0.01, *** *p* < 0.001 vs. UVA-treated cells.

**Figure 6 ijms-23-06960-f006:**
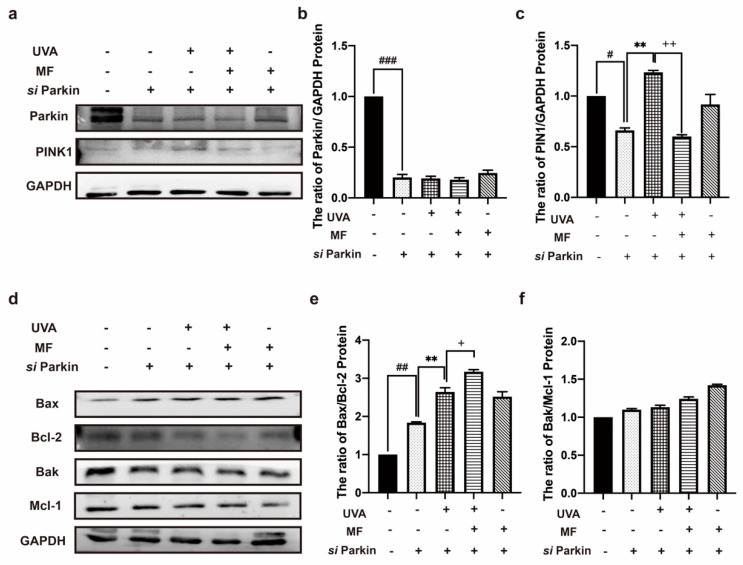
Alteration of Parkin expression can regulate apoptosis in HFF cells after UVA irradiation. (**a**–**c**) Levels of mitophagy proteins Parkin, PINK1, measured by western blot in response to Parkin knockdown. (**d**–**f**) Levels of apoptosis-related proteins Bax/Bcl−2 and Bak/Mcl−1 measured by western blot in response to Parkin knockdown. Data are presented as mean ± SD, n = 3. ^#^
*p* < 0.05, ^##^
*p* <0.01, ^###^
*p* < 0.001 vs. control cells, ** *p* < 0.01 vs. UVA-treated cells, ^+^
*p* < 0.05, ^++^
*p* < 0.01, vs. metformin co-treated cells.

**Figure 7 ijms-23-06960-f007:**
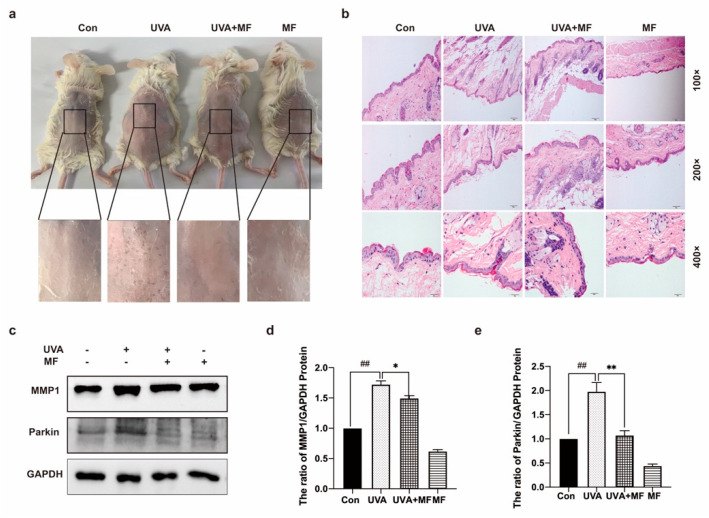
Metformin can improve UVA-induced skin photoaging in mice. (**a**) Representative photographs of mice from control group, UVA-irradiated group, the experimental group receiving metformin, and metformin only group. (**b**) Histological sections of mouse skin. Groups are divided according to the following: control, UVA, metformin co-treated, metformin. The images were acquired using light microscopy (top to bottom 100×, 200×, 400× magnification). (**c**–**e**) The protein levels of MMP1 and Parkin were measured by western blot. Data are presented as mean ± SD, n = 3. ^##^
*p* < 0.01 vs. control group; * *p* < 0.05, ** *p* < 0.01 vs. UVA-treated group.

**Figure 8 ijms-23-06960-f008:**
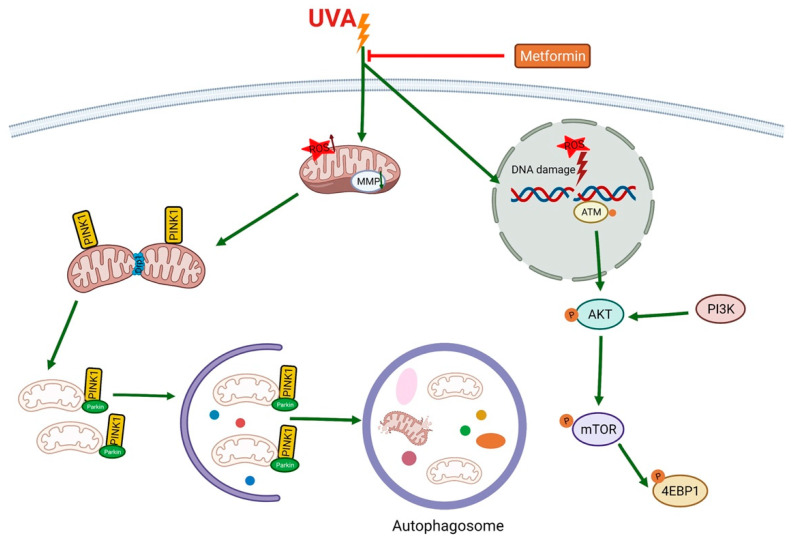
UVA increases mitochondrial ROS levels and decreases mitochondrial membrane potential, leading to mitophagy, and induces DNA damage and activates PI3K/AKT/mTOR signaling pathway, which causes cell senescence, while metformin suppresses mitochondrial damage, mitophagy, and PI3K/AKT/mTOR signaling pathway response to UVA-induced photoaging.

## Data Availability

Data sharing is not applicable.

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
