# Peer review of "Metformin Attenuates UVA-Induced Skin Photoaging by Suppressing Mitophagy and the PI3K/AKT/mTOR Pathway"

_ijms, 2022, doi:10.3390/ijms23136960_

Round 1
Reviewer 1 Report
The article by Dr. Yu and the group elaborates on the role of Metformin in UVA-Induced Skin Photoaging and sheds light on the mechanism of the phenomenon. This is a well-written manuscript that follows the hypothesis of the work, though a few things need to be addressed before it is ready for acceptance. They are as follows:
1. In Figure 3a, the authors need to show the p-S6K level in western blot, it will be used as another confirmation of the mTORC1 downstream effector. Since the p-4EBP1 western blot is not clean, we need another target as a confirmation.
2. Similarly, for fig 4 c, NRF2 level must be chacked- a major marker for the antioxidant pathway. HO western blot is not that clean enough to make any conclusion. So please run NRF2 with proper antibody and detect the level in Western blot analysis.
3. Similar concept was postulated before- like the antidiabetic drug being used for cancer therapy- in a couple of research papers PMID: 26323019 and PMID: 30820523, authors should mention these in the discussion section. It will signify the current work from this manuscript with different combinations of treatments for therapeutic purposes.
4. Another point that must be discussed as a future aspect of this study is whether oncogenes play any role in UVA-induced skin photoaging. Since it has been discussed that oncogenic KRAS upregulates mitophagy (PMID: 33870211) and we know that TGF beta oncogenic signaling pathway is involved in photoaging (PMID: 33358021). Authors should add a few lines discussing their perspective on the topic of the role of oncogenes like KRAS, BRAF, etc in UVA-induced Skin Photoaging. This addition will be helpful for the readers to get a broader aspect of this study.
Author Response
Point 1: In Figure 3a, the authors need to show the p-S6K level in western blot, it will be used as another confirmation of the mTORC1 downstream effector. Since the p-4EBP1 western blot is not clean, we need another target as a confirmation.
Response 1: Thanks for reviewer’s suggestion. We are so sorry that we could not make a supplement of p-pS6K at present. Because COVID-19 is serious in our city, we are not allowed to enter the laboratory for experiment. If the western blot of p-pS6K is required, we need a few weeks. In our result, we detected the expression of p-PI3K, p-AKT, p-mTOR, and p-4EBP1. All these results suggested a significant trend of the PI3K/AKT/mTOR signaling pathway. In addition, our manuscript not only dealed with PI3K/AKT/mTOR pathway, but with mitophagy. So, we think the result of p-pS6K is not required.
Point 2: Similarly, for fig 4 c, NRF2 level must be chacked- a major marker for the antioxidant pathway. HO western blot is not that clean enough to make any conclusion. So please run NRF2 with proper antibody and detect the level in Western blot analysis.
Response 2: Before we submitted the manuscript, we have examined the total Nrf2 expression in HFFs cells treated with UVA and/or MF, but not Nrf2 expression in the nucleus. Following the reviewer's suggestion, we have supplied the results of NRF2 in Figure 4c,f.
Point 3: Similar concept was postulated before- like the antidiabetic drug being used for cancer therapy- in a couple of research papers PMID: 26323019 and PMID: 30820523, authors should mention these in the discussion section. It will signify the current work from this manuscript with different combinations of treatments for therapeutic purposes.
Response 3: According to the reviewer’s suggestion, we have added the content in the discussion. In this study, we found that there was a new role of MF in alleviating chronic photoaging. This is in addition to anti-tumor and hypoglycemic effects.
Point 4: Another point that must be discussed as a future aspect of this study is whether oncogenes play any role in UVA-induced skin photoaging. Since it has been discussed that oncogenic KRAS upregulates mitophagy (PMID: 33870211) and we know that TGF beta oncogenic signaling pathway is involved in photoaging (PMID: 33358021). Authors should add a few lines discussing their perspective on the topic of the role of oncogenes like KRAS, BRAF, etc in UVA-induced Skin Photoaging. This addition will be helpful for the readers to get a broader aspect of this study.
Response 4: We have added the discussion according to the editor’s suggestion. At present, there are many oncogenes involved in skin photoaging, such as TGFβ, KRAS, BRAF, but their mechanism of action is still unclear. In future, the role of oncogenes in skin photoaging needs to be further investigated.
Reviewer 2 Report
-
However, the mechanism by which MF inhibits cell senescence in chronic skin damage induced by UVA. UVA?
-
In this study, we used UVA-induced human foreskin fibroblasts (HFFs) as an in vitro models. Incorrect sentence.
-
and UVA-induced skin photoaging in KM mice as an in vivo model to investigate the potential skin protective mechanism of MF. KM? Erratic use of abbreviations. Rewrite abstract with proper usage of abbreviations.
-
Metformin is depicted as metformin and Mf at some places. Maintain consistency
-
The connectivity seems an issue in the Introduction section. It seems as teh information is just written without proper connectivity and hence it needs to be improved in this context.
-
Results are well written with good demonstration of images.
-
The effect of cellular cytotoxicity on cell survival was assessed using the MTT assay. CIte a proper reference.
-
Add a paragraph regarding future perspective of the study.

Author Response
Point 1: However, the mechanism by which MF inhibits cell senescence in chronic skin damage induced by UVA. UVA?
Response 1: UV irradiation is divided into three distinct spectral areas, including UVC (200–280 nm), UVB (280–315 nm), and UVA (315–400 nm). Wavelengths <290 nm are blocked by stratospheric ozone; so there is no natural exposure to UVC. UVB penetrates the ozone layer and constitutes 5%–10% of the terrestrial solar UV radiation. Radiation in the UVA range is by far the most abundant solar UV radiation (>90%) that reaches the surface of earth. UVA penetrates human skin more efficiently than UVB. UVA has the longest wavelength and penetrates to the levels of the upper dermis in human skin, and UVB only penetrates down to the statum basale. So, chronic skin damage is the result of accumulation of damage to the dermis part of the deeper layers of the skin induced by UVA.
Point 2: In this study, we used UVA-induced human foreskin fibroblasts (HFFs) as an in vitro models. Incorrect sentence.
Response 2: We have modified this sentence in the abstract. In this study, human foreskin fibroblasts (HFFs) treated with UVA was used as an in vitro model.
Point 3: and UVA-induced skin photoaging in KM mice as an in vivo model to investigate the potential skin protective mechanism of MF. KM? Erratic use of abbreviations. Rewrite abstract with proper usage of abbreviations.
Response 3: We have modified KM mice to Kunming mice and checked the manuscript carefully.
Point 4: Metformin is depicted as metformin and Mf at some places. Maintain consistency
Response 4: Thanks for the reviewer’s kindness. We have checked and modified all metformin in the text to MF, except for the subtitles and legends in the results, as this was requested by the editor.
Point 5: The connectivity seems an issue in the Introduction section. It seems as teh information is just written without proper connectivity and hence it needs to be improved in this context.
Response 5: We have made appropriate modifications based on the reviewer’s suggestions.
Point 6: Results are well written with good demonstration of images.
Response 6: Thanks for the reviewer’s kindness.
Point 7: The effect of cellular cytotoxicity on cell survival was assessed using the MTT assay. CIte a proper reference.
Response 7: According to the reviewer’s advice, we have added the proper reference for the MTT assay in the materials and methods.
Point 8: Add a paragraph regarding future perspective of the study.
Response 8: According to the reviewer’s suggestion, we have added a new paragraph regarding future perspective of the study.
Round 2
Reviewer 1 Report
All concerns have been addressed, ready for acceptance.